# A Critical Look at Linus Pauling’s Influence on the Understanding of Chemical Bonding

**DOI:** 10.3390/molecules26154695

**Published:** 2021-08-03

**Authors:** Sudip Pan, Gernot Frenking

**Affiliations:** 1Jiangsu National Synergetic Innovation Center for Advanced Materials, School of Chemistry and Molecular Engineering, Institute of Advanced Synthesis, Nanjing Tech University, Nanjing 211816, China; sudip.chem88@gmail.com; 2Fachbereich Chemie, Philipps-Universität Marburg, Hans-Meerwein-Str. 4, 35037 Marburg, Germany

**Keywords:** chemical bond, valence bond theory, resonance, molecular orbital theory, Lewis electron-pair model

## Abstract

The influence of Linus Pauling on the understanding of chemical bonding is critically examined. Pauling deserves credit for presenting a connection between the quantum theoretical description of chemical bonding and Gilbert Lewis’s classical bonding model of localized electron pair bonds for a wide range of chemistry. Using the concept of resonance that he introduced, he was able to present a consistent description of chemical bonding for molecules, metals, and ionic crystals which was used by many chemists and subsequently found its way into chemistry textbooks. However, his one-sided restriction to the valence bond method and his rejection of the molecular orbital approach hindered further development of chemical bonding theory for a while and his close association of the heuristic Lewis binding model with the quantum chemical VB approach led to misleading ideas until today.

It is hardly possible to adequately acknowledge Linus Pauling’s achievements in so many areas of science and humanity in a single issue of a journal, let alone in an article, even if one focuses on his epoch-making contributions to chemistry. Shortly after his death in 1994, four books with biographies had already been published [1,2,3,4,5] on the life and impact of the man who was the only one to date to receive the Nobel Prize twice undivided. His description of chemical bonding on a quantum theoretical basis in conjunction with the electron pair model of Gilbert Lewis can still be found in many publications today, in which the structures and bonding situations of molecules are presented in a way that goes back to the ideas and conceptions of Pauling. His book “*The Nature of the Chemical Bond*”, first published in 1939 with the third edition in 1960, was undoubtedly the most influential work in the field, greatly inspiring and shaping the understanding of chemical bonding for generations of chemists [6]. A cursory glance at today’s chemistry textbooks and chemical literature shows that his view of chemical bonding has left a lasting impression that is still visibly reflected in modern research.

This article serves to appreciate the importance of Linus Pauling’s work for the understanding of chemical bonding by indicating and discussing its limitations and weaknesses. Where there is a lot of light, there is also shadow. This was already expressed by two fellow giants of chemical bonding theory. Robert Mulliken, who received the Noble Prize in 1966 for the development of the molecular orbital (MO) theory, wrote in his memoir “*As a master salesman and showman, Linus persuaded chemists all over the world to think of typical molecular structures in terms of the valence bond method*” [7]. When Mulliken was asked by a science historian for his opinion about Pauling’s contribution to our understanding of chemical bonding, his reply was “*He set it back fifteen years*” [8]. A similar statement is found at the end of the autobiography of Erich Hückel, who wrote in 1975 about Linus Pauling’s book: “*He succeeded in stopping the progress of science for 20 years* (Original in German, translated by the author)” [9]. To understand the background of this harsh criticism, it is helpful to look at the early beginnings of the physical understanding and modeling of chemical bonding using quantum theory and Pauling’s influence on its development.

In 1927, Walter Heitler and Fritz London published in Zurich a quantum-theoretical study of the chemical bond in H_2_, which can be regarded as the birth of the physical understanding of the covalent chemical bond, previously one of the great mysteries of the natural sciences [10]. They used the newly developed quantum theory of Werner Heisenberg [11] and Erwin Schrödinger [12], which provided a physical framework for the unusual behavior of, and forces between, elementary particles that could not be properly explained by classical physics. Heitler and London were young postdocs in the laboratory of Schrödinger and it belongs to the curiosities of scientific history that their mentor refused to be an author on the paper, because he was not interested in chemistry. At the same time, Linus Pauling was another postdoctoral fellow in Schrödinger’s group with the help of a Guggenheim fellowship, which he took advantage of to learn about quantum theory from Nils Bohr in Copenhagen, Schrödinger in Zürich, and Arnold Sommerfeld in Munich, where he also met with Werner Heisenberg. But unlike most physicists, Pauling had a deep knowledge of chemistry and X-ray crystallography and he was familiar with the electron pair model of chemical bonding introduced by Lewis [13,14] that was further developed by Irving Langmuir [15,16,17,18]. What set Pauling apart was the combination of excellent knowledge of chemistry with learning quantum theory, which was at that time in a nascent state, coupled with great ambition, determination, and an unshakable self-confidence. Pauling realized that the quantum theoretical method used by Heitler and London could be directly associated with Lewis’ electron pair model, which lacked a physical foundation.

The Heitler–London (HL) approach, now referred to as the valence bond (VB) method, uses localized two-center product functions λ_a_λ_b_ to build the VB wave function Ψ_0_^VB^, which is expressed as the sum of all possible combinations of the product pairs of atomic orbitals λ. There are several variants of VB methods known [19,20] but the essential features are the same. The basic VB expansion has the electron-sharing covalent term (λ_a_ − λ_b_) (“Heitler–London (HL)” term) and the two ionic terms (λ_a_|^−^ λ_b_^+^) and (λ_a_^+^ λ_b_|^−^):Ψ_0_^VB^ = Σ c_1_ (λ_a_ − λ_b_) + Σ c_2_ (λ_a_|^−^ λ_b_^+^) + Σ c_3_ (λ_a_^+^ λ_b_|^−^) + Mix(1)

The coefficients c_n_ of the pair functions λ_a_λ_b_ that are occupied by two electrons may be used directly to assign a covalent or ionic character to a chemical bond A-B. A full VB calculation aiming at the correct energy of the molecule considers not only the pure covalent and ionic terms in Equation (1) but also their mixing contributions, Mix, to the total energy, which is named as resonance energy. The concept of resonance and resonance energy plays a central role in Pauling’s approach to chemical bonding. A more detailed description of the VB approach is given in the literature [19,20]

It is useful to point out the central piece of information about the physical nature of chemical bonding that was already identified in the work of Heitler and London with the statement: “…*crucial for the understanding of the possible behaviors between neutral atoms turns out to be a characteristic quantum mechanical vibrational phenomenon*. (Original in German, translated by the author)” [10]. It is the description of the electrons as wavefunctions but not as charges, which leads to a physical understanding of the formation of a chemical bond. The essential features are outlined in Scheme 1, which gives a mathematically simplified expression of the bond formation in H_2_. A more detailed discussion is found in a recent review article, which gives also an account of the history of the electron pair bonding model of Lewis [21].

Scheme 1 describes first the classical approach for the interaction between two hydrogen atoms where the electrons are presented by their electronic charge distribution ρ(**r**) as the basic entity. The sum of the charge distributions of the two hydrogen atoms, ρ(H_a_) and ρ(H_b_), leads to an approximate charge distribution of the hydrogen molecule, ρ(H_2_) (Equation (2)). The associated curve for the purely electrostatic interaction energy E_elstat_ (Equations (3) and (4)) shows only a shallow energy minimum of ~10 kcal/mol at a rather long H–H distance. This is shown in Figure 1 which is adapted from the original work of Heitler and London [10] where the electrostatic interaction energy is termed as E_11_.

The quantum theoretical approach uses the wave function ψ(**r**) as the fundamental quantity of electrons. The relationship between the wave function ψ(**r**) and the charge density ρ(**r**) is given by the square, ρ(**r**) = [ψ(**r**)]^2^ (Equation (4)), which when rearranged gives ψ(**r**) = ±√ρ(**r**). Thus, the wave function ψ(**r**) carries a ± phase factor which is crucial for the understanding of the electronic structure of the molecule. The quantum theoretical ansatz for H_2_ uses the wave functions of the hydrogen atoms to construct the molecular wave function ψ(H_2_) (Equation (6)), which gives two solutions. This is fundamentally different to the classical approach, which results from a quantum theoretical treatment of the electrons. The equation for the quantum mechanical electronic charge distribution [ψ(H_2_)]^2^ (Equation (7)) leads, after insertion of the binomial (Equation (6)), to two solutions (Equation (8)), which represent the charge distributions of the bonding and antibonding states of H_2_. A comparison of Equation (8) with (2) reveals that the quantum theoretical Equation (8) contains a completely new contribution called “interference” or “resonance”, which is either added or subtracted. The associated energy expressions in Equations (9) and (10) exhibit the curves for E_α_ and E_β_ in Figure 1 A classical description of the electron-electron interaction leads to repulsion due to Coulomb’s law, while a quantum theoretical description of the electron–electron interaction using wave functions leads to attraction due to their interference. *Covalent bonding is a quantum theoretical interference phenomenon.*

It is important to recognize the difference between the mathematical ansatz for the VB wavefunction that gives the total energy and the assignment of the HL and ionic terms to covalent and ionic bonding, which is based on Pauling’s proposal. Whereas the VB ansatz (Equation (1)) provides a valid description of the electronic structure of the molecule, the identification of (λ_a_ − λ_b_) with covalent bonding and (λ_a_|^−^ λ_b_^+^)/(λ_a_^+^ λ_b_|^−^) with ionic bonding is only a model. It correlates directly with the Lewis electron pair model based on classical physics and is appealing because of its simplicity, but is misleading for understanding covalent bonding. This becomes already obvious when dihydrogen is considered. A full VB calculation of H_2_ shows that ~90% of the bond dissociation energy (BDE) comes from the HL term H–H but ~10% comes from the ionic terms, although the bond is fully covalent. The contributions of the ionic terms become larger and may even be dominant when polar bonds such as in LiF are considered. But the diatomic LiF has a polar covalent bond, where the accumulated charge in the bonding region is shifted toward fluorine, rather than an ionic bond. In contrast, solid LiF has ionic bonds coming from the instantaneous attraction between the charged species Li^+^ and F^−^ in the ionic crystal, which have much longer Li–F distances and negligible overlap of the atomic valence orbitals compared with diatomic LiF.

Shortly after the publication of Heitler and London, an alternative approach to the VB method for the calculation of molecular electronic structures, called the MO method, was introduced by Mulliken [22,23] and Friedrich Hund [24,25,26,27,28]. Mathematically, MO theory rests on the product of sums while VB theory is based on the sum of products. The final results of MO and VB calculations are the same when all terms are considered, but the MO wave function Ψ_0_^MO^ is generally delocalized over the entire molecule and appears incompatible with Lewis’ localized electron pair model of chemical bonding, which has proved to be very powerful in explaining the structures and bonding situation of molecules. This was one reason why Pauling strongly opposed and often ridiculed the MO method. His biting comments were feared. When Charles Coulson published in 1952 the textbook on quantum chemistry “*Valence*” [29] where he compared the essential features of the VB and MO theories, it received a very hostile review by Pauling [30]. Another early pioneer of the molecular electronic structure was Nevil Sidgwick, who appears to be forgotten in the literature. He wrote in 1927 [31] and 1933 [32] two textbooks about valency, where he suggested to use the symbol of an arrow for a bond A→B where both bonding electrons come from the same atom A [33,34,35,36]. Such a bonding situation was introduced by Lewis and is the basis for the definition of acids and bases named after him [14,37]. Sidgwick proposed the name “co-ordinate bond” [33] for what is now commonly called a dative bond. It is the basis of the Dewar–Chatt–Duncanson (DCD) model for transition metal complexes, which goes back to a 1951 proposal by Michael Dewar for bonding in Zeise’s salt [38] that was systematically extended by Chatt and Duncanson to other transition metal compounds [39].

In retrospect, Pauling’s strong opposition to MO theory is difficult to understand, because the advantages of MO theory over the VB approach were obvious even before the advent of computers. In molecules, electrons form a shell structure with defined quantum numbers and energy values analogous to atoms, by which molecular orbitals and atomic orbitals are described. This allowed Hund to explain the electron spectra of diatomic molecules, for which he also introduced among others the now-commonly used symmetry symbols σ, π, δ [27] indicating the change in sign of an MO when mirrored at one or two mirror planes, which are nowadays often mistakenly identified with the bond multiplicity. Lennard-Jones showed already in 1929 [40] that MO theory easily explains the X^3^Σ_g_^−^ electronic triplet ground state of O_2_, which had been a puzzle for Lewis and is a failure of his electron-pair model [13,14]. Perhaps, the most important information about molecules offered by molecular orbitals is the spatial symmetry of MOs. This was recognized by Fukui in the 1950s when he presented the mathematical basis of frontier orbital theory for describing the structure and reactivity of molecules [41,42,43,44]. The importance of orbital symmetry for understanding chemical reactions as described by the Woodward–Hoffmann rules for pericyclic reactions [45] has been demonstrated in organic chemistry in several seminal works [46,47,48,49,50,51,52]. The electronic structure of transition metal complexes can easily be explained with the help of MO correlation diagrams where the spatial symmetry of the orbitals plays a crucial role [53,54]. The same is true for compounds of heavier atoms of the first octal row of the periodic system such as PF_5_ and SF_6_, where the apparent failure of the octet rule is corrected when the symmetry of the orbitals is taken into account [55,56]. The trend of the chemical bonds in the series of diatomic molecules of the first octal row from Li_2_ to F_2_ with the nonexistence of a genuine chemical bond in Be_2_ and the triplet ground states of B_2_ and O_2_ can be readily explained with the symmetry of the orbitals [57].

Figure 2 shows the correlation diagram of the orbitals for diatomic molecules E_2_ (E = Li − F) including symmetry allowed hybridization. The occupation of the MOs following the Aufbau principle provides a straightforward explanation for the electronic ground state of E_2_ for the first octal-row atoms. The energy difference between the 1π_u_ MO and the 3σ_g_^+^ orbital is rather small and the ordering changes for some heavier diatomic systems [53,57]. A special case is C_2_, which has a X^1^Σ_g_^+^ ground state [58] suggesting, according to the orbital occupation, a formal double bond with two π bonds but no σ bond. But the small HOMO–LUMO gap makes C_2_ a multiconfigurational molecule where the contribution of excited configurations increases the bond order. A few years ago, it was even suggested that C_2_ has a quadruple bond, which is stronger than the carbon–carbon triple bond in acetylene HC≡CH although the latter has a significantly shorter CC bond [59,60,61,62]. The suggestion was based on VB calculations of C_2_, which led to controversies [63,64,65,66] with some workers who proposed on the basis of MO calculations and other VB studies a bond order that is between two and three [67,68,69,70,71,72]. But the picture of a quadruple bond in C_2_ was supported by other workers using a variety of methods [73,74,75,76]. It became clear that neither the simple Lewis image with a double bond |C=C| nor that with a quadruple bond with three strong and one weak component C≡C correctly represent the bonding situation in C_2_. There was agreement that all four valence electrons of carbon are involved in the binding interactions, but the strength of the two σ bonds was disputed. The controversy was recently solved by an experimental study of the photoelectron spectra of the C_2_^−^ anion. Using a high-resolution photoelectron imaging spectrometer, the results show that the dominant contribution to the dicarbon bonding in C_2_ involves a double-bonded configuration, with 2π bonds and no accompanying σ bond [77].

However, it is important to realize that the decomposition of the total wave function into individual MOs is only a model, and that the correlation between MOs and physical and chemical properties is a helpful approach not a description of physical reality, even though the energy levels of the MOs due to Koopman’s theorem [78] are often in surprisingly good agreement with the ionization energies of a molecule measured by photoelectron spectroscopy [79]. The strength of the MO model is the spatial symmetry of the orbitals, which provides important information for understanding and predicting the structure and reactivity of molecules.

The term “symmetry” is completely missing in the subject index still in the latest edition from 1960 of “*The Nature of the Chemical Bond*” and so is the name of Kenichi Fukui [6]. It seems that Pauling did not realize the fundamental information on the relevance of symmetry that was provided by MO theory. The important work on chemical bonding in diatomic and polyatomic molecules by Friedrich Hund published in 1932 is not even mentioned [80,81,82]. Pauling had extensive knowledge of the synthetic chemistry of his time, but he ignored the information about chemical bonding that was emerging from molecular spectroscopy. His rigid adherence to the model of localized chemical bonding led to a misleading equalization of Lewis’s electron pair model with the quantum theoretical VB approach of Heitler and London, which can still be traced today. The important distinction between a model of chemical bonding and the physical basis of the electronic interaction leading to chemical bonding was obscured by Pauling’s work. The idea that a bonding model is not right or wrong, it is more or less useful was aptly expressed by Michael Dewar who wrote in 1984: “*The only criterion of a model is usefulness, not its ‘truth’*” [83]. This is to be contrasted with a statement about the physical nature of chemical bonding made by Mulliken in 1932: “*Attempts to regard a molecule as consisting of specific atoms or ionic units held together by discrete numbers of bonding electrons or electron pairs are considered as more or less meaningless, except as an approximation in special cases or as a method of calculation*.” [84]. It may be added that such an approach is extremely useful for describing the structure and reactivity of molecules, but it must be clear that this is a useful model and not the physical reality of the electronic structure and chemical bonds.

The physical nature of the chemical bond has been the topic of many theoretical studies following the pioneering work of Heitler and London in 1927 and many other ground breaking studies of early quantum chemistry by Pauling [85], Lennard-Jones [40], Slater [86], Bethe and Fermi [87], Hückel [88], Hellmann [89], Mulliken [22,23,84], Hund [24,25,26,27,28,80,81,82] and others appeared. The present understanding of the physical origin of covalent bonding was reviewed by Werner Kutzelnigg [90] and more recently by Michael Schmidt, Joseph Ivanic and Klaus Ruedenberg [91], who published in 1962 a seminal paper on the topic [92]. The understanding and interpretation of the chemical bond in terms of either the wavefunction ψ(**r**) or the charge distribution ρ(**r**) is often controversially discussed. This is also caused by the fact that often no distinction is made between the physical mechanism of the interatomic interaction and the final chemical bond which is thereby built. While the former can only be understood with the help of the wave function ψ(**r**), the generated chemical bond and the associated electronic structure can be completely described by the charge distribution ρ(**r**).

It is instructive to examine Pauling’s attempts to describe the bonding situation in molecules where Lewis’ electron pair model fails or encounters problems. The limitation of the electron-pair bonding model comes clearly to the fore in the section about transition metal carbonyl complexes in his book “*The Nature of the Chemical Bond*” where the author describes the metal–CO bonds in terms of resonating structures [93]. The 1960 edition presents a discussion of ferrocene Fe(C_5_H_5_)_2_ for which 560 resonating structures are suggested [94]. This may be compared with a single MO correlation diagram, which directly describes the crucial features of the metal–ligand bonding [53,95]. But it does not take large transition metal complexes to encounter the limitation of the electron pair model, it becomes obvious when the nonexistence of doubly bonded Be_2_ or the (^3^Σ_g_^−^) triplet ground state of O_2_ is considered. Dioxygen may straightforwardly be written with a Lewis structure that fulfills the octet rule and has a double bond 
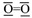
, but this is in conflict with its triplet ground state. Lewis already struggled with the triplet state of O_2_ in his presentation of the electron pair model, which appeared to be an anomaly for describing the bonding situation in molecules [13,14]. Pauling introduced in 1931 the concept of one-electron and three-electron bonds [96] and he suggested to write O_2_ with the following “structure **A**” (Figure 3).

Since the bond order of a three-electron bond was suggested to be one half of a two-electron bond, the total bond order in structure **A** is two, which agrees with the electronic structure. But structure **A** is nothing more than a posteriori description of the triplet state of O_2_ with the Lewis model, which is extended ad hoc by a three-electron bond. Pauling describes other molecules there [96] and in his book [6], such as OF and Cl_2_^−^ and nitroso complexes of transition metals, that might contain a three-electron bond, and he speculates about the electronic interactions that lead to an energetic preference over normal two-electron bonds. In doing so, he ignores the earlier works by Lennard-Jones in 1929 on the triplet state of O_2_ [40] and by Hund, who provided between 1927 and 1930 a simple explanation for electronic states in the ground and excited states based on MO theory [24,25,26,27,28].

There is one interesting aspect in the history of Pauling’s preference for the VB method and his ignorance of the MO approach. In 1935, he published a paper with G.W. Wheland entitled “*A Quantum Mechanical Discussion of Orientation of Substituents in Aromatic Molecules*” [97]. In the outline of the method it is said “*There are two principal methods available for the quantum mechanical treatment of molecular structure, the valence bond method and the molecular orbital method. In this paper we shall make use of the latter, since it is simpler in form and is more easily adapted to quantitative calculations*.” which shows that Pauling knew that the MO method was simpler and better suited for quantitative calculations than the VB method, but he apparently rejected the former because it seemed incompatible with the Lewis model.

The electronic state of dioxygen has recently been calculated with modern methods of VB theory, which were further developed by Cooper, Gerratt et al. [20,98] and lately by Shaik, Hiberty and co-workers [19], who also introduced a variety of VB-based models for a wide range of molecular structures and reactivities [99]. Not surprisingly, the advanced VB calculations correctly give the (X^3^Σ_g_^−^) triplet ground state of O_2_ [100]. VB theory and MO theory are two alternative quantum chemical methods that approach the same numerical results at higher levels of theory, albeit at very different costs. But along with the results of the VB calculations, it was claimed that this disproves the myth of the failure of the VB theory in describing the triplet ground state of O_2_ [101]. The assertion is based on an incorrect equalization of VB theory and the Lewis model, which can be traced back to Pauling. This is clearly expressed in a recent review by Shaik [102] where he writes: “*Valence Bond (VB) theory has been born thrice. The first time in 1916* [1] *when Lewis postulated that the quantum unit of chemical bonding is an electron pair*….” quoting ref. [13]. The erroneous equalization of a classical model with a quantum theoretical treatment of chemical bonding was repeated in other places [103,104] and it is the source of misleading statements. VB theory is a quantum chemical approach for the mathematical description of the electronic structure of molecules, which was introduced by Heitler and London in 1927 [10]. The paper by Lewis in 1916 [13] suggested that the chemical bond may be identified with a pair of electrons. It has no reference to quantum theory, which was actually disliked by Lewis [105]. Pauling connected VB theory with the Lewis electron pair model, which helped chemists think of chemical bonding in terms of the simple Lewis model, but now combined with the complicated quantum theoretical approach to molecular electronic structure. But the Lewis model and the VB method are completely different approaches and equating them can lead to deceptive conclusions, as shown in the following.

Following Pauling’s suggestion, VB calculations using Equation (1) identifies (λ_a_ − λ_b_) with covalent bonding and (λ_a_|^−^ λ_b_^+^)/(λ_a_^+^ λ_b_|^−^) with ionic bonding. But there are electron-rich molecules like F_2_ where neither of the two terms give the largest contribution to the binding energy, which is rather provided by the resonance (Mix) term. A related situation is found when F_2_ is subject of MO calculations, where the Hartree–Fock approximation gives a negative value for the bond energy, which is corrected by including correlation energy. It is well known that molecules with high electron densities require higher angular momentum wave functions in order to give an accurate description of the electronic structure, but this does not indicate a different type of bonding. The finding that the resonance term of the VB calculation of F_2_ delivers the largest part of the BDE creates a fundamental problem for the orthodox interpretation of VB results. It was suggested that the resonance (Mix) term of a VB calculation identifies a previously unrecognized new type of chemical bond called a “charge-shift bond” (CSB), which is distinctly different from covalent and ionic bonds [106,107,108,109]. Since the VB energy terms in Equation (1) can be mapped on equivalent MO calculations, the energy contribution of the Mix term from a VB calculation can be estimated from an analogous MO calculation of the same molecule. This was taken as evidence that CSB bonding is also found in MO theory [110].

But a detailed analysis of the physical origin of the bonding in diatomic molecules B_2_, C_2_, N_2_, O_2_ and F_2_ by Ruedenberg and co-workers showed that the physical nature of the interatomic interactions is the same, and that all diatomics are bonded by covalent interactions [74]. The driving force of the covalent bond is the interference of the wavefunctions. The common feature is the lowering of the kinetic energy of the valence electrons, which are delocalized over several atoms. Each diatomic molecule of the first octal row exhibits peculiarities, such as Li_2_, where the removal of an electron from the bonding pair leads to strengthening rather than weakening of the bond in Li_2_^+^ [57]. At the other end of the octal series, the high electron density in F_2_ exhibits a particular electronic structure and a comparatively weak bond, weaker than in Cl_2_, which is one of the rare cases where a bond between atoms of the first octal series is weaker than the analogous bond of the second series. But there is no physical basis for the claim that F_2_ has a new type of chemical bond rather than a covalent bond. The CSB can be proposed as a new model, but then there is the question of its usefulness beyond the introduction of a new name [111,112,113].

The ill-fated muddling of a model and the physical nature of chemical bonding that started from Pauling’s linking of VB theory with the Lewis picture can be traced to the recent past. In 2019, Liu et al. reported the observation of the anion [NaBH_3_]^−^ which was claimed to possess an Na^−^→BH_3_ dative bond [114]. This was questioned by Pan and Frenking who suggested instead a classical polar covalent bond Na−BH_3_^−^ [115], which was rejected by Liu et al. providing arguments in favor of a dative bond [116]. In a following series of papers, authors introduced different types of sodium–boron bonds for the molecule. Foroutan-Nejad suggested “*ionic-enforced covalency*” for the [Na-BH_3_]^−^ interaction [117] while Salvador et al. proposed another bonding category namely a spin-polarized bond [118]. More recently, Radenkovic et al. presented VB calculations which were claimed to be “*solving the conundrum*” by showing that the molecule has neither a dative nor a polar-covalent bond but rather “*a unique combination of coulombic stabilization with Charge-Shift Bonding character*” which leads to a “*combination of (major) dipole-dipole electrostatic interaction and (secondary) resonant one-electron bonding mechanism*” [119]. Finally, Pino-Rios et al. suggested an “*one-electron Na**●B Bond*” in [NaBH_3_]^−^ but they also clearly stated “*The physical nature of the bond is not a puzzle, but rather, it comes from the interference of the atoms’ wave functions. What makes the NaBH_3_^−^ bond unusual is the difficulty of describing it with standard bonding models*” [120]. This hits the point, but it may not be the end of discovering still new bonding categories, which has become a fashionable hobby these days. It is reminiscent of the appearance of the number of new types of aromaticity, which reached 43 in 2016 [121].

Ted Goertzel, co-author of one of the biographies of Linus Pauling [3], wrote: “When his brilliance as a scientific innovator declined with age, he fell more and more into his second intellectual style [becoming emotionally committed to his ideas and seeking out evidence to support them]. In his later years, his combativeness and defensiveness increasingly triumphed over his brilliance and creativity” [122]. This profession-related danger exists for scientists who discover or believe they have discovered something new, and who are so obsessed with their (apparent) findings that they ignore other aspects and arguments that call their discoveries into question. The only cure for this disease is to remain open to valid arguments against one’s conclusion, which is the duty of helpful opponents.

What then remains of the Pauling legacy in the field of chemical bonding in the form of VB calculations and the associated localized bonding models in the form of resonance between Lewis structures? Quantitative VB calculations no longer have any utility except for very small molecules because the computational efficiency of MO calculations is dramatically better. The difference in computational cost became even greater with the advent of modern density functional theory (DFT) calculations, which can be viewed as parameterized Hartree–Fock calculations where the additional terms in the operator lead to the Kohn–Sham equations. DFT calculations are the main workhorse of today’s computational chemistry, complemented by correlated ab initio calculations to obtain highly accurate numerical results. VB calculations could be performed with DFT optimized geometries to achieve a localized bonding description. But the MO and DFT calculations can be directly transformed into a localized picture of the bonding situation that recovers the Lewis bonding model as well. This could either be done via unitary transformation of the delocalized canonical MOs to a localized MOs or by using the popular NBO (natural bond orbital) method of Frank Weinhold [123]. No VB calculation is required to obtain a localized electron pair description of the chemical bonds in a molecule. However, one must be aware that in any transformation of the total wave function of a molecule into a localized Lewis bonding model, there is a subjective factor that comes from the transformation algorithm designed and coded by the author. Chemists trained and accustomed to the model of resonance introduced by Pauling may still find it useful to use VB calculations, but they forgo all the information given by the symmetry of the wave function [124,125]. However, it is fair to say that problems like the homolytic bond breaking of electron-pair bonds are more easily dealt with using VB calculations, which remains an important approach in the toolbox of quantum chemical methods [99,100,101,102,103,104,126,127]. The VB and MO methods have been compared and discussed in several papers, which shed further light on the two approaches [128,129,130,131,132,133,134].

Linus Pauling deserves credit for introducing a connection between the quantum theoretical description of chemical bonding and Gilbert Lewis’s classical bonding model of localized electron pair bonds for a wide range of chemistry. Using the concept of resonance that he introduced, he was able to present a consistent description of chemical bonding for molecules, metals, and ionic crystals that was used by many chemists and found its way into chemistry textbooks. His one-sided restriction to the VB method and his rejection of the MO approach hindered further development of chemical bonding theory for a while. His close association of the heuristic Lewis binding model with the quantum chemical VB approach tempted some people to equate the unrelated quantities, leading to misleading notions to this day. Isaac Newton once wrote “If I could see further, it was because I was standing on the shoulders of giants.” Pauling clearly belongs to the group of giants of chemical bonding. But standing on his shoulders means that we are supposed to be able to see further than he could. Simply repeating the words of the giants would mean that we have made no progress. This also includes a critical examination of the results and conclusions of the pioneers.

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
