# Peer review of "A Critical Look at Linus Pauling’s Influence on the Understanding of Chemical Bonding"

_molecules, 2021, doi:10.3390/molecules26154695_

Round 1

Reviewer 1 Report

From the very first reading of the paper I have very ambiguous feelings. To be honest I have gone through this manuscript couple of times being probably more and more confused about its scientific value. But after having some distance I have ultimately enjoyed the content.

Clearly, the paper is well written and from my perspective it is a great guide through the history of the scientific efforts focused on the theoretical description of chemical bonding with special emphasis on the point of view served by Linus Pauling. There is rather interesting discussion related to the VB theory, in particular how this theory has been received by scientific audience. The introduction into the topic of the VB theory is rather shallow, the authors probably assume the reader is familiar with the concepts, which perhaps is correct attitude. However, for the completeness of the picture it would be valuable to make it a bit more involved, or at least provide explicitly relevant references. The Essay form is probably relevant for providing some sort of introductory paragraph focused more on basics, it would really improve the didactic part of the paper. I suppose that the Essay form is aiming to have one, to some extent at least. I have enjoyed some information that I was not aware of, e.g. the fact that it symmetry of the MOs was not properly appreciated by Pauling. From our current perspective it is rather difficult to disregard this fact. What is clearly surprising to me is a quite strong statement at the end. The authors clearly controvert the Pauling’s achievements, which is perhaps justified from the current perspective. But, just by having in mind the historical context and the fact that the VB theory in a beautiful way connects quantum chemistry and relatively basic chemical principles, I would temper a bit the strength of the statement. But this is my purely subjective opinion, also related to the fact on my personal scientific way the VB theory allowed me to find a bridge between chemistry and mathematical description of molecular species.

The Scheme 1 seems to be incomplete, there are clear references to Eqns. 1-4, but there are missing, obviously this should be fixed.

All together my humble recommendation would be to publish the paper with a kind request to the readers to take into account the above statements and modify relevant paragraphs accordingly.
